# Beneficial Effects of *Astragalus membranaceus* (Fisch.) Bunge Extract in Controlling Inflammatory Response and Preventing Asthma Features

**DOI:** 10.3390/ijms241310954

**Published:** 2023-06-30

**Authors:** Danilo D’Avino, Ida Cerqua, Hammad Ullah, Michele Spinelli, Rita Di Matteo, Elisabetta Granato, Raffaele Capasso, Lucianna Maruccio, Armando Ialenti, Maria Daglia, Fiorentina Roviezzo, Antonietta Rossi

**Affiliations:** 1Department of Pharmacy, School of Medicine and Surgery, University of Naples Federico II, 80131 Naples, Italy; danilo.davino@unina.it (D.D.); ida.cerqua@unina.it (I.C.); hammad.ullah@unina.it (H.U.); rit.dimatteo@gmail.com (R.D.M.); ialenti@unina.it (A.I.); maria.daglia@unina.it (M.D.); 2Department of Chemical Sciences, University of Naples Federico II, 80100 Naples, Italy; michele.spinelli@unina.it; 3Department of Agricultural Sciences, University of Naples Federico II, 80055 Portici, Italy; rafcapas@unina.it; 4Department of Veterinary Medicine and Animal Production, University of Naples Federico II, 80137 Naples, Italy; lucianna.maruccio@unina.it

**Keywords:** *Astragalus membranaceus* extract, inflammation, macrophages, asthma, allergen, atopy

## Abstract

*Astragalus membranaceus* (Fisch.) Bunge root is used as herbal medicine for its immunomodulating activities in Chinese medicine. Recently, beneficial properties of *A. membranaceus* on allergic diseases have been proposed. Here we investigated the role of a commercial extract of *A. membranaceus*, standardized to 16% polysaccharides, in regulating the immune-inflammatory response in vitro and in vivo and its therapeutic application in asthma. *A. membranaceus* extract inhibited prostaglandin E_2_ and leukotriene C_4_ production in stimulated J774 and peritoneal macrophages, respectively. The extract also reduced interlukin-1β, tumor necrosis factor-α, and nitrite production, affecting inducible nitric oxide synthase expression. In vivo experiments confirmed the anti-inflammatory properties of *A. membranaceus*, as evident by a reduction in zymosan-induced peritoneal cellular infiltration and pro-inflammatory mediator production. The efficacy of *A. membranaceus* extract in modulating the immune response was confirmed in a model of allergic airway inflammation. Extracts improve lung function by inhibiting airway hyperresponsiveness, airway remodeling, and fibrosis. Its anti-asthmatic effects were further sustained by inhibition of the sensitization process, as indicated by a reduction of ovalbumin-induced IgE levels and the mounting of a Th2 immune response. In conclusion, our data demonstrate the anti-inflammatory properties of the commercial extract of *A. membranaceus* and its beneficial effects on asthma feature development.

## 1. Introduction

*Astragalus membranaceus* (Fisch.) Bunge is a well-known medicinal species from the *Astragalus* genus, and its root (*Astragali radix*) is used in herbal medicine. In traditional Chinese medicine, *A. membranaceus* has been used to treat weakness, wounds, anemia, fever, chronic fatigue, loss of appetite, uterine bleeding, and uterine prolapse [1]. It is used to treat chronic phlegmatic disorders and general gastrointestinal disturbances, including stomach ulcers and diarrhea [1]. *A. membranaceus*, recorded in the pharmacopoeias of Europe, China, the United States, Japan, and Korea [2,3,4,5,6], contains various chemical constituents, including triterpenoid saponins, flavonoids, and flavonoid glycosides [1,7]. Because of its abundant biological activities, in recent years it has been applied to functional foods. Many studies suggest that this herb may act as an immunoregulator [1,8,9]. Indeed, in vitro and in vivo experiments show that *A. membranaceus* may enhance the body’s natural defense mechanisms [1]. It has been demonstrated that *A. membranaceus* polysaccharides stimulate macrophage functions [8] and regulate the expression of cytokines such as interleukin (IL)-1, and IL-6, as well as the production of nitric oxide (NO) [9]. Macrophages play a central role in the immune/inflammatory response through the production of several pro-inflammatory mediators such as arachidonic acid (AA) metabolites (leukotrienes (LT) and prostaglandins (PG)), NO, and cytokines. These mediators are responsible for typical hallmarks of inflammation (i.e., oedema, increased vascular permeability, and cell infiltration), and LTs are crucial mediators of allergic diseases such as bronchial asthma, allergic rhinitis, and urticaria [10,11].

Recently, beneficial properties of *Astragalus* for allergic diseases have been proposed. However, the molecular mechanisms involved have not been clarified. Here we investigated the immune modulatory action of *A. membranaceus* by using a commercial root extract of *A. membranaceus* standardized to 16% polysaccharides. For this purpose, we investigated the anti-inflammatory effects of *A. membranaceus* in vitro and in vivo and evaluated its therapeutic application in a model of experimental asthma.

## 2. Results

### 2.1. Astragalus membranaceus Extract Chromatographic Analysis

This study used a dry commercial hydroalcoholic extract produced from the root of *A. membranaceus*, standardized to contain 16% polysaccharides. The sample was analyzed by liquid chromatography-mass spectrometry (LC-MS)/MS in multiple reaction monitoring mode (MRM) by using a mass spectrometry targeted method based on the analysis of 85 molecules to characterize the polyphenolic profile (Figure 1).

The analysis was carried out by HPLC-MS/MS mass spectral analysis of the phytocomplex. The MRM method led to the identification and quantification of 37 metabolites, as shown in Table 1, by using the external standard method, as described in the Material and Methods section. The extract has a very complex metabolic profile, containing all flavonoid subclasses: flavonols, flavones, flavanones, anthocyanidins, flavanols and isoflavones. Among the most commonly detected compounds, three isoflavonoids (biochanin, formononetin, and glycitein) were identified. Moreover, the extract contains diosmetin and hydroxycinnamic acid derivatives (i.e., chlorogenic acid, caffeoyliquinic acid lactone, and ferulic acid).

### 2.2. Astragalus membranaceus Extract Inhibits Macrophage Activation

#### 2.2.1. Effect of *Astragalus membranaceus* Extract on Cell Viability

J774 macrophages were treated with *A. membranaceus* extract (0.25–1 mg/mL, 24 h) and cell viability was assessed using a 3-(4,5-dimethylthiazol-2-yl)-2,5-diphenyltetrazolium bromide (MTT) assay. No significant effects of *A. membranaceus* extract were observed on cell viability at all concentrations tested (Figure 2).

#### 2.2.2. *Astragalus membranaceus* Extract Inhibits LPS-Induced PG Production in J774 Macrophages

Macrophages are an important source of AA-derived lipid mediators such as PGs, which display a wide variety of functions in the initiation and regulation of inflammatory response. Stimulation of J774 macrophages with lipopolysaccharide (LPS, 10 μg/mL; 24 h) induced a significant increase in PGE_2_ generation in comparison to unstimulated control cells (Figure 3A). In the presence of the *A. membranaceus* extract (0.25–1 mg/mL, 2 h before LPS) a significant and concentration-dependent inhibition of PGE_2_ production was observed (Figure 3A), with an IC_50_ of 0.51 mg/mL. The inhibitory effect of *A. membranaceus* extract on PGE_2_ production in LPS-stimulated cells was reverted by exogenous AA added (Figure 3B). In the presence of exogenous AA, only the higher concentration of *A. membranaceus* extract (1 mg/mL) exhibited weak inhibition of LPS-induced PGE_2_ production (Figure 3B). In addition, *A. membranaceus* extract significantly reduced the production of 6-keto-PGF_1α_ (a stable metabolite of prostacyclin, induced by LPS) in a concentration-dependent manner (Figure 3C), with an IC_50_ of 0.76 mg/mL. The effect of *A. membranaceus* extract on PG biosynthesis was not correlated to the inhibition of cyclooxygenase (COX)-2 expression. Indeed, LPS stimulation for 24 h induced a significant increase of COX-2 expression in J774 macrophages that wasn’t affected by *A. membranaceus* extract pre-treatment (Figure 3D, Appendix A).

#### 2.2.3. *Astragalus membranaceus* Extract Inhibits LPS-Induced Nitrite Production in J774 Macrophages

To evaluate whether the *A. membranaceus* extract was able to reduce NO production, the Griess assay was used to measure nitrite levels, the stable end-product of NO, in supernatants of the murine macrophage J774 stimulated with LPS (10 µg/mL). The stimulation of cells with LPS for 24 h induced a marked increase in nitrite production in the cell medium with respect to unstimulated macrophages (Figure 4A). The pre-treatment of macrophages with *A. membranaceus* extract (0.25–1 mg/mL, 2 h before LPS stimulation) significantly decreased nitrite production in the cell medium in a concentration-dependent manner, with an IC_50_ value of 0.76 mg/mL (Figure 4A). Next, we evaluated whether the inhibitory effect of *A. membranaceus* on LPS-induced nitrite production was mediated by the regulation of inducible nitric oxide synthase (iNOS) protein expression. As shown in Figure 4B, the treatment of cells with LPS (10 mg/mL, for 24 h) induced a significant increase in iNOS protein expression with respect to unstimulated cells (control). When cells were pre-treated with the *A. membranaceus* extract (0.5–1 mg/mL), iNOS protein expression was inhibited with respect to LPS-stimulated cells (Figure 4B, Appendix A).

#### 2.2.4. *Astragalus membranaceus* Extract Reduces LPS-Induced IL-1β and TNF-α Production in J774 Macrophages

To further investigate the in vitro anti-inflammatory effects of *A. membranaceus* extract, the levels of pro-inflammatory cytokines such as IL-1β and TNF-α were evaluated in LPS-stimulated J774 macrophages. The stimulation of cells with LPS for 24 h induced a significant increase in IL-1β and TNF-α in comparison to unstimulated cells. As shown in Figure 4, the extract reduced both IL-1β (Figure 4C) and TNF-α (Figure 4D) production induced by LPS stimulation, with an IC50 value of 1 mg/mL.

#### 2.2.5. *Astragalus membranaceus* Extract Reduces LTC_4_ Production in A23187-Activated Peritoneal Macrophages

Peritoneal macrophages are an important source of LT, which plays a pivotal role in inflammation. The effect of the commercial extract of *A. membranaceus* on LT production was evaluated. Stimulation of mouse peritoneal macrophages with A23187 (0.5 µg/mL, 60 min) induced a significant increase in LTC_4_ generation compared to unstimulated cells. Pre-treatment of cells with the commercial AM extract (0.25–1 mg/mL, 30 min before A23187) caused a significant and concentration-dependent inhibition of LTC_4_ production (Figure 5).

### 2.3. Astragalus membranaceus Extract Attenuates Inflammation in Murine Peritonitis

To evaluate the in vivo effects of *A. membranaceus* extract, an experimental model of acute inflammation, i.e., zymosan-induced peritonitis, was used. Resident peritoneal macrophages are the leukocyte subtype mainly involved in the early production of pro-inflammatory mediators during peritonitis [12]. Zymosan treatment induces extravasation of plasma containing first-line chemotactic factors (e.g., LTB_4_) that mediate subsequent neutrophil influx and their accumulation in the inflamed tissue [12]. In particular, the peak of leukocyte infiltration in the peritoneal cavity occurs 4 h after zymosan injection [12]. Indeed, we observed a significant increase in cell infiltration as several cells migrated into the peritoneal cavity with respect to control animals (Figure 6A). Cell recruitment was reduced by *A. membranaceus* extract pre-treatment, with the best results at a dose of 100 mg/kg (Figure 6A). Accordingly, the extract significantly alleviated the zymosan-induced LTC_4_ (Figure 6B) and LTB_4_ (Figure 6C). *A. membranaceus* extract also showed in vivo anti-inflammatory effects by inhibiting zymosan-induced PGE_2_ (Figure 6D), 6-ketoPGF_1α_ (Figure 6E), nitrite/nitrate (NOx, Figure 6F), IL-1β (Figure 6G), and TNF-α (Figure 6H) in the peritoneal exudates of zymosan-treated mice.

### 2.4. Astragalus membranaceus Extract Exhibits Beneficial Effect on Asthma Features

The effect of *A. membranaceus* extract in an experimental model of asthma was assessed by exposing the mice to the extract during the phase of allergen sensitization. To achieve this aim, BALB/c mice were pre-treated with *A. membranaceus* 30 min before ovalbumin (OVA) injection on days 0 and 7 and killed after 21 days to evaluate bronchial hyperreactivity, pulmonary inflammation, and injury, as well as plasma IgE, pulmonary Th2 cytokine, and LTB_4_ production. OVA sensitization induced a significant increase in bronchial reactivity to carbachol (Figure 7A) as well as bronchial relaxation in response to salbutamol (Figure 7B) in sham animals. Treatment of mice with *A. membranaceus* fully reversed OVA-induced bronchial hyperreactivity to carbachol (Figure 7A) and restored the adrenergic bronchial relaxation induced by salbutamol (Figure 7B). The effects on airway reactivity were well correlated with airway remodeling evaluated as smooth muscle actin (SMA) expression (Figure 7C,D). Indeed, OVA-sensitization induced an increase in α-SMA expression that was reduced by the AM pre-treatment (Figure 7C,D). Histological analysis with hematoxylin-eosin staining (Figure 7E) shows a cuboidal and regular bronchial epithelium in the lung of sham mice (Figure 7E). In the peri-bronchovascular interstitium of the lung of sham mice, the connective fibers are regular and minimally dense (Figure 7E). Following the allergen sensitization, the bronchial epithelium showed structural changes, and the epithelial cells became cylindrical with intracellular spaces. OVA exposure also induced changes in the peri-bronchovascular section with the recruitment of inflammatory cells (Figure 7E,F) and the thickening of the connective fibers (Figure 7E). The pre-treatment with *A. membranaceus* induced a partial recovery in the structure of the bronchial epithelium (Figure 7E). Indeed, the epithelial cells returned cuboidal, but some intracellular spaces remained. A partial effect of *A. membranaceus* was also evident in the reduction of pro-inflammatory cells (Figure 7E,F) in the peri-bronchovascular section, but the connective fibers were reduced as well as in the control. In addition, OVA sensitization induced an increase in the goblet cell number as shown by periodic acid-Schiff (PAS) staining compared to the sham group (Figure 7G), resulting in a mucus production increase. *A. membranaceus* pre-treatment had no effect on mucus production (Figure 7G).

OVA sensitization also induced extensive peri-bronchial (b) and peri-vascular (v) fibrillar collagen accumulations compared to the sham group (Figure 8A(a,b), B), resulting in an increase in stiffness in the lung. In the same way, the primary collagen, evaluated with Masson’s trichrome, increased in OVA-sensitized mice (Figure 8C(d,e), D) compared to the control group, resulting in a damaged basement membrane. The *A. membranaceus* pre-treatment significantly reduced the primary collagen (Figure 8C(f),D) but not the fibrillar one (Figure 8A(c),B).

The beneficial properties of *A. membranaceus* on lung function were associated with their effects on sensitization mechanisms. Indeed, *A. membranaceus* significantly reduced plasma immunoglobulin (Ig)E levels (Figure 9A) as well as pulmonary T-helper type 2 cytokines, such as IL-13 and IL-4 (Figure 9B,C), induced by OVA sensitization. In addition, as observed in an in vitro study and in zymosan-induced peritonitis, AM extract reduced LT levels in sensitized mice (Figure 9D).

## 3. Discussion

Here we demonstrate that a dry commercial hydroalcoholic extract produced from the root of *A. membranaceus*, standardized to 16% polysaccharides, exerts anti-inflammatory effects in in vitro and in vivo models with multi-target action. *A. membranaceus* extract (1) reduces PGs, NO, IL-1β and TNF-α in LPS-stimulated J774 macrophages; (2) decreases LTC_4_ in A23187-stimulated peritoneal macrophages; (3) inhibits in vivo inflammatory reactions in zymosan-induced peritonitis; and (4) exhibits beneficial effects on asthma features (Figure 10).

Inflammatory diseases are on the list of 24 priority disorders that the World Health Organization (WHO) published in 2013, characterized as high-burden conditions for which the currently available treatment is inadequate. Indeed, despite anti-inflammatory drug discovery being very intensive, inflammatory diseases remain among the most serious health burdens, and the medical need for more potent and safe anti-inflammatory drugs remains. Inflammation is a very complex pathophysiological process, involving a myriad of mediators such as AA metabolites (PG and LT), NO, and cytokines. Therefore, pharmacological modulation of a single target tends to generate shunting phenomena that can compromise therapeutic efficacy [13]. A multi-target, polypharmacological approach appears to be more suitable. In this context, natural products containing a complex of bioactive compounds are of great interest for inflammation pharmacotherapy and offer an alternative strategy to classical non-steroidal anti-inflammatory drugs [14,15].

Emerging evidence suggests that *A. membranaceus* has beneficial effects on the immune system, promoting the recovery of patients affected by chemotherapy, radiotherapy [16], or acquired immune deficiency syndrome [17]. In particular, the effects of *A. membranaceus* as an immune enhancer have been partially attributed to its anti-inflammatory and immunomodulatory properties [7]. In this study, we used a commercial *Astragalus membranaceus* root dry extract containing all flavonoid subclasses: flavonols, flavones, flavanones, flavanols, isoflavones, and anthocyanidins. Among the most commonly detected compounds, three isoflavonoids (biochanin, formononetin, and glycitein) with many beneficial properties such as antioxidant [18], anti-inflammatory, and cardioprotective activities [19] were identified. Moreover, the extract contains diosmetin, a flavone that is known to exert cytoprotective and antibacterial properties [20]. Another chemical class represented in the extract is that of hydroxycinnamic acid derivatives (i.e., chlorogenic acid, caffeoyliquinic acid lactone, and ferulic acid), known for their antioxidant and neuroprotective activity [21]. The composition of *A. membranaceus* was found to be very interesting, with all the classes of antioxidant molecules [22].

In our conditions, the commercial *A. membranaceus* extract showed anti-inflammatory activity through the inhibition of pro-inflammatory mediator production. This extract inhibited the production of PGE_2_ in LPS-stimulated J774 macrophages at noncytotoxic concentrations. The first step of PG biosynthesis is the liberation of AA from cell membrane phospholipids via the hydrolysis of the sn-2 bond by the phospholipase A_2_ enzyme. AA is then oxygenated by COX to form PGG_2_ and subsequently reduced to yield the unstable intermediate, PGH_2_. Subsequently, PGH_2_ is the substrate for various specific enzymes (PG synthases) that produce more stable prostanoids, including PGE_2_, PGI_2_, PGD_2_, PGF_2α_, and thromboxane A_2_ [23]. The inducible isoenzyme microsomal PGE synthase (mPGES-1) is functionally coupled to COX-2, the primary COX controlling PG synthesis in response to inflammation, and converts PGH_2_ into PGE_2_, the major prostanoid regulating inflammation [24]. Our data suggest that the inhibitory effects of *A. membranaceus* on PGE_2_ biosynthesis are related to an action on AA release rather than on COX-2 or mPGES enzymes expression. This data contrasts with what was published by Ryu et al. which have demonstrated that an *Astragali radix* extract inhibited LPS-induced COX-2 expression in RAW264.7 macrophages at very low concentrations (0.1 μg/mL) [25]. These differences could be explained by the different compositions of the two extracts.

To evaluate the possible effects of *A. membranaceus* extract on mPGES-1, the levels of 6-keto-PGF_1α_, a stable metabolite of PGI_2_, were evaluated. Our data showed that the levels of 6-keto-PGF_1α_ were decreased in J774 macrophages following *A. membranaceus* treatment, suggesting an effect upstream of the synthesis of PG. The multiple effects of *A. membranaceus* on the PG pathway were also evidenced by others using an *A. membranaceus* sub-fraction [26], containing five isoflavonoids and eight saponins. In perfect tune, the concentration-dependent inhibition of PGE_2_ production observed with the *A. membranaceus* sub-fraction didn’t correlate to COX-2 inhibition [26]. Moreover, Ryu et al. maintained that the inhibition of COX-2 expression was due to a substantial reduction of IL-1β production. In our experimental conditions, we observed a significative inhibition of IL-1β only at the highest concentration tested. The hypothesized mechanism of action of *A. membranaceus* extract on AA release was confirmed by an inhibitory effect of the extract on LT production in A23187-stimulated peritoneal macrophages. This could also be attributed to the presence of formononetin and biochanin, the most abundant flavonoids in the extract. As already reported, formononetin inhibits AA release in LPS-stimulated RAW264.7 [27].

During inflammation, in response to various inflammatory stimuli, macrophages generate micromolar amounts of NO that are highly toxic to the surrounding cells and tissues [28]. Consequently, a therapeutic anti-inflammatory strategy is to suppress NO production by inhibiting iNOS expression. Our results demonstrate that LPS-induced NO production in J774 cells is susceptible to inhibition by the commercial *A. membranaceus* extract through iNOS protein expression reduction. These data are in perfect tune with what was observed by others using *A. membranaceus* extracts and subfractions, although they obtained effects at lower concentrations [25,26].

The HPLC analysis of the commercial *A. membranaceus* extract showed the presence of several secondary metabolites such as polyphenolic flavonoids, which have anti-inflammatory and immunomodulatory effects. It has been demonstrated that the most present flavonoids, formonetin, biochanin A, and glycitein, may inhibit mitogen-activated protein kinase activation and pro-inflammatory mediator production in several experimental models of inflammation [29,30,31], suggesting their involvement in the anti-inflammatory effects of our commercial *A. membranaceus* extract. Interestingly, the in vitro beneficial effects of *A. membranaceus* extract on macrophage functions were confirmed in the zymosan-induced peritonitis model. Macrophages are the cell subtype mainly involved in early pro-inflammatory mediator production during peritonitis [12]. Specifically, in our experimental conditions, peritoneal macrophages are responsible for the production of LT and the onset of the inflammatory process, with a peak of inflammatory cell influx at 4 h post zymosan [12]. *A. membranaceus* extract reduced the LT production and cell infiltration in zymosan-treated mice. Furthermore, consistent with this effect, *A. membranaceus* extract decreased the levels of pro-inflammatory mediators such as PG, NO, and cytokines (IL-1β and TNF-α).

In traditional Chinese medicine, the decoction of *Astragali radix* is used to treat allergic diseases and chronic cough [32]. Indeed, it has been demonstrated in a randomized, double-blind, placebo-controlled trial that an herbal-mineral complex containing an extract of the root of *A. membranaceus* improved the symptoms and quality of life in patients with allergic rhinitis [32]. In addition, an *Astragali radix* decoction greatly improves the symptoms of allergic airway remodeling in experimental models [33]. Despite a generally accepted beneficial effect in boosting the body’s general vitality and strengthening resistance to allergens, it has rarely been investigated for its effect in preventing asthma features. In this study, we investigated the effects of *A. membranaceus* extract on allergen-sensitized BALB/c mice, expressing a Th2-high immunophenotype that mirrors human atopy and having susceptibility to develop some of the main cardinal asthma signs such as elevated IgE levels, airway inflammation, and hyperresponsiveness. Our data demonstrated that the commercial *A. membranaceus* extract improved airway function, lung remodeling, and inflammation induced by allergen sensitization in mice. These protective effects involved the action of the extract on mounting the Th2 immune response underlying sensitization. Indeed, *A. membranaceus* extract reduced plasma IgE levels and Th2 cytokine production (IL-4 and IL-13) induced by OVA sensitization and pro-inflammatory LTB_4_ production. Overall, the data shows that the extract not only has a therapeutic effect in the acute phase of asthma but also has efficacy in preventing asthma feature development by acting on the sensitization mechanisms.

In conclusion, our data demonstrate that the commercial extract of *A. membranaceus* displays immunomodulatory effects through the in vitro and in vivo inhibition of multiple pathways in macrophages as well as a protective effect during allergen sensitization (Figure 10).

## 4. Materials and Methods

### 4.1. Astragalus membranaceus Extract

A commercial extract obtained using as extraction solvent water-ethanol (70/30 *v*/*v*) of *Astragalus membranaceus* Moench root (16% polysaccharides dry extract, with a drug/extract ratio of 10/1) was used (A.C.E.F. Azienda Chimica e Farmaceutica, Fiorenzuola d’Arda, Italy). All other reagents and fine chemicals were obtained from Sigma-Aldrich (Milan, Italy).

### 4.2. Astragalus membranaceus Extract Analysis by HPLC-MS/MS in Multiple Reaction Monitoring

The dry extract was suspended in a mix of water and methanol 50:50 (*v*/*v*). The extraction of polyphenols was conducted under stirring for three hours in dark conditions. After, 1 mL of solution was centrifuged at 10,000 rpm for 10 min and the supernatant was filtered through 0.45 µm polytetrafluoroethylene syringe filters and then directly transferred into High-Performance Liquid Chromatography (HPLC) auto sampler. One microliter of supernatant was analyzed by using an AB-sciex 5500 QTRAP^®^ system with an HPLC chromatography system Exion LC™. The mobile phase was generated by mixing eluent A (0.1% formic acid in water) and eluent B (0.1% formic acid in acetonitrile) and the flow rate was 0.200 mL min^−1^. The chromatographic gradient was from 20% to 90% B in 4 min, held for 2 min, then returned to 20% B in 2 min. Tandem mass spectrometry was performed using a Turbo V TM ion source operated in negative ion mode, and the multiple reaction monitoring (MRM) mode was used for the selected analytes. Appendix A provide a list of precursor ions, product ions, collision energy, and declustering potential parameters. The extracted mass chromatogram peaks of metabolites were integrated using Skyline software version 22.2 for data processing.

### 4.3. J774 Cell Culture

The murine monocyte/macrophage J774 cell line was obtained from the American Type Culture Collection (ATTC TIB 67). The cell line was grown in adhesion in Dulbecco’s modified Eagles medium (DMEM) supplemented with glutamine (2 mM, Aurogene, Rome, Italy), Hepes (25 mM, Aurogene, Rome, Italy), penicillin (100 U/mL, Aurogene, Rome, Italy), streptomycin (100 μg/mL, Aurogene, Rome, Italy), fetal bovine serum (FBS, 10%, Aurogene Rome, Italy) and sodium pyruvate (1.2%, Aurogene, Rome, Italy) (DMEM completed). The cells were plated at a density of 1 × 10^6^ cells in 75 cm^2^ culture flasks and maintained at 37 °C under 5% CO_2_ in a humidified incubator until 90% confluence. The culture medium was changed every 2 days. Before a confluent monolayer appeared, a sub-culturing cell process was carried out. Cells were pre-treated for 2 h in the absence or presence of test compound (0.25, 0.5, 0.75, and 1 mg/mL) and then stimulated for 24 h with LPS from *Escherichia coli*. After 24 h of incubation with LPS, the supernatants were collected for the PGE_2_, 6-keto-PGF_1α_, nitrite, IL-1β, and TNF-α measurement. In another set of experiments, the cells were stimulated with LPS for 24 hrs to induce COX-2, then pre-treated for 2 h with *A. membranaceus* extract and further incubated for 30 min with AA (15 µM) [34]. At the end of incubation, the supernatants were collected for the measurement of PGE_2_ and 6-keto-PGF_1α_ levels (Cayman Chemical, Vinci-Biochem, Vinci, Italy) by Enzyme-Linked Immunosorbent Assay (ELISA) assay. The nitrite concentration in the samples was measured by the Griess reaction, by adding 100 μL of Griess reagent (0.1% naphthylethylenediamide dihydrochloride in H_2_O and 1% sulphanilamide in 5% concentrated H_2_PO_4_; vol. 1:1; Sigma Aldrich, Milan, Italy) to 100 μL samples. The optical density at 540 nm (OD_540_) was measured immediately after Griess reagent addition, using an ELISA microplate reader (Thermo Scientific, Multiskan GO, Milan, Italy). Nitrite concentration was calculated by comparison with OD_540_ of standard solutions of sodium nitrite prepared in a culture medium. IL-1β and TNF-α levels were measured with commercially available ELISA kits according to the manufacturer’s instructions (R&D Systems, Aurogene, Rome, Italy).

#### 4.3.1. Cell Viability

Cell respiration, an indicator of cell viability, was assessed by the mitochondrial-dependent reduction of MTT (Sigma Aldrich, Milan, Italy) to formazan. Cells were plated to a seeding density of 1.0 × 10^5^ in 96 multiwells. After stimulation with the test compound for 24 h, cells were incubated in 96-well plates with MTT (0.2 mg/mL), for 1 h. The culture medium was removed by aspiration and the cells were lysed in dimethyl sulfoxide (DMSO) (0.1 mL). The extent of reduction of MTT to formazan within cells was quantified by the measurement of OD_550_ [35].

#### 4.3.2. Western Blot Analysis

The analysis of COX-2 and iNOS in J774 macrophages was performed on whole cell lysates. After stimulation with LPS for 24 h, cells were washed with cold PBS, collected by scraping, and centrifuged at 8000 rpm for 5 min at 4 °C. Pellets were lysed by syringing with RIPA buffer (Trizma Base, NaCl, EDTA 100 mM, Na-deossicolate 10% and 10% Nonidet P-40) completed with orthovanadate activated 200 mM and complete protease inhibitor cocktail (Sigma Aldrich) and centrifuged at 12,000 rpm for 10 min at 4 °C. The supernatants were collected and protein concentration in cell lysates was determined by Bio-Rad Protein Assay (Bio-Rad, Milan, Italy). Equal amounts of protein (50 μg) were mixed with gel loading buffer (50 mM Tris, 10% SDS, 10% glycerol, 10% 2-mercaptoethanol, and 2 mg/mL of bromophenol) in a ratio of 4:1, boiled for 5 min. Each sample was loaded and electrophoresed on a 10% SDS–polyacrylamide gel. The proteins were transferred onto nitrocellulose membranes (0.2 μm nitrocellulose membrane, Trans-Blot^®^ TurboTM, Transfer Pack, Bio-Rad Laboratories, Milan, Italy). The membranes were blocked with 0.1% phosphate-buffered solution (PBS)-Tween containing 5% non-fat dry milk. After the blocking, the membranes were incubated with the relative primary antibody overnight at 4 °C. Mouse monoclonal antibodies anti COX-2, iNOS (BD Transduction Laboratories, Milan, Italy) were diluted 1:1000 in 0.1% PBS-Tween, 5% non-fat dry milk; mouse monoclonal antibody anti β-actin (Santa Cruz Biotechnology, Dallas, TX, USA) was diluted 1:1000 in 0.1% PBS-Tween, 5% non-fat dry milk. After the incubation, the membranes were washed three times with 0.1% PBS-Tween and were incubated for 2 h at room temperature with horseradish peroxidase-conjugated anti-mouse secondary antibody (Santa Cruz Biotechnology Dallas, TX, USA) diluted 1:2000 in 0.1% PBS-Tween containing 5% non-fat dry milk. The membranes were washed and protein bands were detected by an enhanced chemiluminescence system (ChemiDoc, Bio-Rad, Milan, Italy). Densitometric analysis was performed with Image Lab software (version 6.1; Bio-Rad Laboratories, Milan, Italy).

### 4.4. Animals

Male CD1 mice (33−39 g body weight, 8 weeks of age, Charles River Laboratories; Calco, Italy) and female BALB/c mice (20 g body weight, 8 weeks of age, Charles River Laboratories, Calco, Italy) were fed with standard rodent chow and water and acclimated for 4 days at a 12 h light and 12 h dark schedule in a constant air-conditioned environment (21 ± 2 °C). Mice were randomly assigned to groups, and experiments were carried out during the light phase. Experimental procedures were conducted in conformity with Italian (D.L. 26/2014) and European (directive 2010/63/EU) regulations on the protection of animals used for scientific purposes and approved by the Italian Ministry.

### 4.5. Resident Peritoneal Macrophages

Resident peritoneal macrophages (PM) were obtained by lavage of the peritoneal cavity of CD1 mice with 7 mL of cold Dulbecco’s modified Eagle’s medium (DMEM) with heparin (5 U/mL). After centrifugation (550× *g* 4 °C for 10 min), PM were seeded in 24-well plates at a density of 5 × 10^5^ cells/mL and allowed to adhere at 37 °C in 5% CO_2_ for 2 h, non-adherent cells were removed by washing with sterile PBS afterward. PM were stimulated with A23187 (0.5 µg/mL) for 60 min [36] after pre-treatment for 30 min with *A. membranaceus* extract (0.25, 0.5, 0.75 and 1 mg/mL) or vehicle (DMSO, 0.5%). Incubation media were assayed for LTC_4_ by ELISA, according to the manufacturer’s instructions (Cayman Chemical, Vinci-Biochem, Vinci, Italy).

### 4.6. Zymosan-Induced Peritonitis

Male CD-1 mice were pre-treated i.p. with *A. membranaceus* extract (100 and 200 mg/kg) or vehicle (0.5 mL, DMSO 2% in saline) 30 min before zymosan (2 mg/mL in saline, i.p., 0.5 mL, Sigma-Aldrich). SHAM group received only saline (0.5 mL). Mice were sacrificed by inhalation of CO_2_ after 30 min or 4 h to analyze peritoneal LTC_4_ (30 min), LTB_4_, cell infiltration, PGE_2_, 6-keto-PGF_1α_, nitrite/nitrate (NOx), IL-1β and TNF-α peritoneal exudate levels. Peritoneal exudates were collected and centrifugated, and cells were counted in exudates after trypan blue staining. Levels of LTB_4_, LTC_4_, PGE_2,_ 6-keto-PGF_1α_ (Cayman Chemical, Vinci-Biochem, Vinci, Italy) and IL-1β and TNF-α (R&D Systems, Aurogene, Rome, Italy) were quantified in the exudate by ELISA according to the manufacturer’s instructions. Measurements of nitrite and nitrate (NOx) were based on the reduction of nitrate to nitrite by cadmium [34] and the subsequent determination of nitrite by the Griess reaction. The reduction of nitrate to nitrite was performed in a microplate: 40 µL of STB (75% NH_4_Cl 0.49 M and 25% of Na_2_B_4_O_7_ 0.06 M) and 115 µL of nitrate standard curves or samples were pipetted in each well. Cadmium granules (2–2.5 g) were rinsed three times with deionized distilled water and then they were added to samples. The microplate was then shaken automatically for 90 min. Subsequently, 155 µL of the mixture from each well was centrifugated, then 100 µL of supernatants were transferred into another microplate. 100 µL of Griess reagent (0.1% naphthylethylenediamide dihydrochloride in H_2_O and 1% sulphanilamide in 5% concentrated H_2_PO_4_; vol. 1:1; Sigma Aldrich, Milan, Italy) was added and absorbance was measured within 10 min in a spectrophotometer at a wavelength of 540 nm.

### 4.7. OVA Sensitization

Female BALB/c mice were treated with 0.4 mL s.c. of a suspension containing 100 µg of OVA from chicken egg white (Sigma-Aldrich, Milan, Italy) absorbed to 3.3 mg of aluminum hydroxide gel (Merck KGaA, Darmstadt, Germany) or saline (sham group) on days 0 and 7 [37,38]. *A. membranaceus* extract (OVA + AM group) (100 mg/kg) or vehicle were administered i.p. 30 min before each OVA administration. Animals were sacrificed on day 21 by an overdose of enflurane, and lungs, bronchi, and blood were collected.

#### 4.7.1. Bronchial Reactivity

OVA-sensitized mice were sacrificed on day 21 by enflurane overdose, exsanguinated, and their lungs were removed. The main bronchi were rapidly dissected and cleaned from fat and connective tissue. Rings of 1–2 mm length were cut and mounted in 3 mL isolated organ baths containing Krebs solution, at 37 °C, oxygenated (95% O_2_ and 5% CO_2_), and connected to an isometric force transducer (type 7006, Ugo Basile, Comerio, Italy) associated to a Powerlab 800 (AD Instruments, Oxford, UK). Rings were initially stretched until a resting tension of 0.5 g was reached and allowed to equilibrate for at least 30 min during which tension was adjusted, when necessary, to 0.5 g and the bathing solution was periodically changed. In each experiment, bronchial rings were previously challenged with carbachol (1 × 10^−6^ M) (Sigma-Aldrich, Milan, Italy) and then a cumulative concentration-response curve of carbachol (1 × 10^−9^ M − 3 × 10^−6^ M) or salbutamol (1 × 10^−8^–3 × 10^−5^ M) was performed to evaluate bronchial reactivity. Results are expressed as dyne per mg tissue [37,38].

#### 4.7.2. Lung Histology

Left lung lobes harvested from mice were fixed in formalin 4%, embedded in paraffin and 7 µm sections were cut. Lung slices were processed to remove paraffin and were rehydrated. Sections were stained with Hematoxylin and Eosin (H&E, Kaltek, Padua, Italy) or PAS (PAS+, Sigma Aldrich, S.I.A.L., Rome, Italy) to evaluate lung structure and mucus production [39]. Other slices were stained with Masson’s Trichrome (Sigma Aldrich, S.I.A.L, Rome, Italy) or Picro-Sirius Red (Abcam, S.I.A.L., Rome, Italy) for the assessment of collagen deposition. Images were acquired by blinded operators using a Leica DFC320 video camera (Leica, Milan, Italy; magnification of 20× for H&E, 40× for PAS+ magnification) connected to a Leica DM RB microscope using the Leica Application Suite software V.4.1.0.

#### 4.7.3. Immunohistochemistry

Lung slides were processed and incubated with 3% hydrogen peroxide (Sigma Aldrich, Milan, Italy) for 15 min to quench the endogenous peroxidase activity. Non-specific interactions were reduced with 2% bovine serum albumin (BSA) in 0.1% PBS-Tween and tissue sections were incubated overnight with anti-actin α-smooth muscle antibody (A5228, Sigma Aldrich, 1:100). After rinsing with PBS 0.01 M, sections were incubated with Peroxidase AffiniPure Goat anti-Rabbit IgG (Jackson ImmunoResearch, 1:500) for 1 h at room temperature. Color development was visualized using 3,3′-Diaminobenzidine Chromogen Solution (SIGMAFAST^TM^-DAB, Merck Millipore, Milan, Italy). Sections were analyzed with a magnification of 20×. Semi-quantitative determination of protein expression was obtained with ImageJ/Fiji software version 2.9.

#### 4.7.4. Measurement of Plasma IgE Levels

Blood was collected by intracardiac puncture using citrate as an anticoagulant. Then plasma was obtained by centrifugation at 800× *g* at 4 °C for 10 min and immediately frozen at −80 °C. Total IgE levels were measured by means of ELISA using matched antibody pairs (BD Biosciences Pharmingen, San Jose, CA, USA).

#### 4.7.5. IL-13, IL-4 and LTB_4_ Levels

Lungs were isolated and homogenized in ice-cold PBS pH 7.4 (100 mg mL^−1^, Sigma Aldrich, Milan, Italy) using a FastPrep tissue homogenizer (2 cycles of 20 s, 6 m/s; MP Biomedicals, DBA, Segrate, Italy). The homogenate was centrifuged (4 °C, 6000× *g*, 10 min). Commercially available ELISA was used to measure the levels of IL-13, IL-4 (R&D Systems, Aurogene, Rome, Italy), and LTB_4_ (Cayman Chemical, Vinci-Biochem, Vinci, Italy) according to the manufacturer’s instructions.

### 4.8. Statistical Analysis

The results are expressed as mean ± S.E.M. of the mean of n observations, where *n* represents the number of animals or the number of experiments performed on different days. In the experiments involving histology, the figures shown are representative of six animals. The results were analyzed by one-way or two-way ANOVA followed by Bonferroni post hoc tests. Post hoc tests were performed only if achieved the *p* < 0.05 level of significance. A *p*-value less than 0.05 was considered significant. All graphs were generated using GraphPad Prism (version 5).

## Figures and Tables

**Figure 1 ijms-24-10954-f001:**
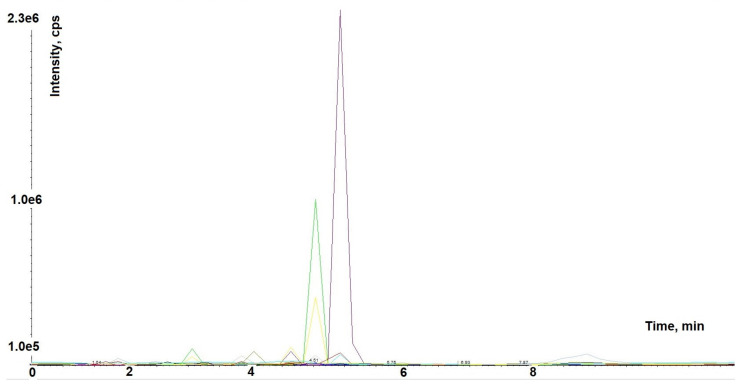
MRM chromatogram of commercial *Astragalus membranaceus* dry extract.

**Figure 2 ijms-24-10954-f002:**
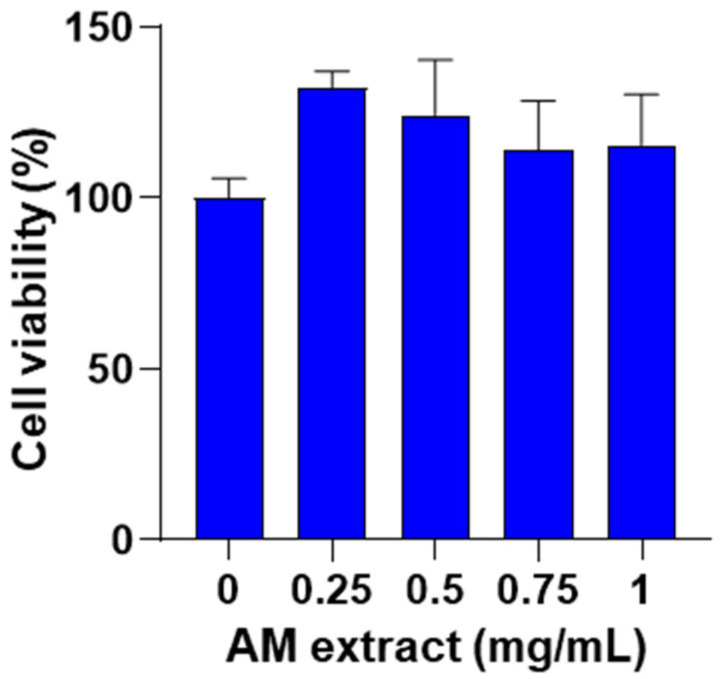
Effect of *A. membranaceus* extract (AM) on cell viability. J774 macrophages were treated for 24 h with AM (0.25–1 mg/mL) and then cell viability was evaluated by the mitochondrial-dependent reduction of MTT to formazan. Values represent means ± S.E.M.; *n* = 3 experiments. Data were analyzed by one-way ANOVA plus Bonferroni.

**Figure 3 ijms-24-10954-f003:**
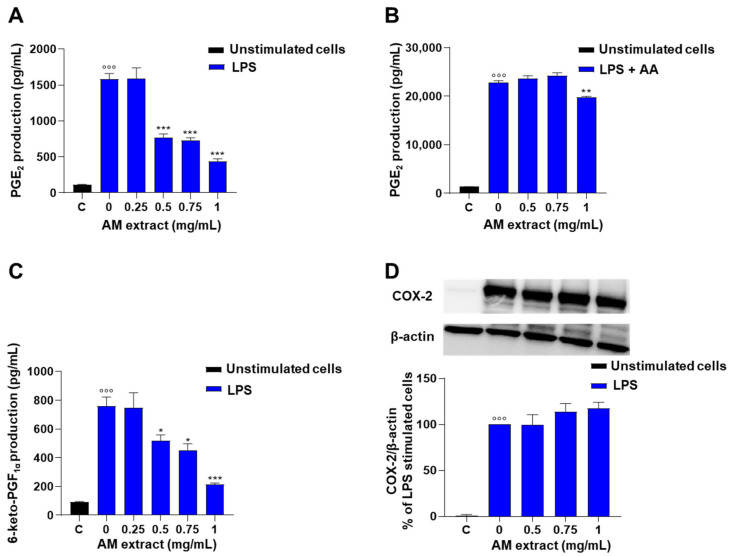
Effect of *A. membranaceus* extract (AM) on PG production in LPS-stimulated murine macrophages: (**A**) J774 cells were pre-treated for 2 h with AM (0.25–1 mg/mL) and then stimulated for 24 h with LPS (10 μg/mL). (**B**) Cells were stimulated, for 24 h, with LPS (10 μg/mL), to induce COX-2, then pre-treated for 2 h with AM and incubated for 30 min with AA (15 µM). The supernatants were collected for the measurement of PGE_2_ levels by ELISA assay. (**C**) J774 cells were pre-treated for 2 h with AM (0.25–1 mg/mL) and then stimulated for 24 h with LPS (10 μg/mL). The supernatants were collected for the measurement of 6-keto-PGF_1α_ levels by ELISA assay. (**D**) Western blots analysis of COX-2 expression in J774 macrophages, pre-treated with AM (0.5–1 mg/mL) for 2 h and exposed to LPS (10 μg/mL) for 24 h. Values represent means ± S.E.M.; *n* = 3 experiments. Data were analyzed by one-way ANOVA plus Bonferroni. Statistical significance is reported as follows °°° *p* < 0.001 vs. unstimulated cells (**C**), * *p* < 0.05 and *** *p* < 0.001 vs. LPS alone (**A**–**C**), ** *p* < 0.01 vs. LPS + AA (**B**).

**Figure 4 ijms-24-10954-f004:**
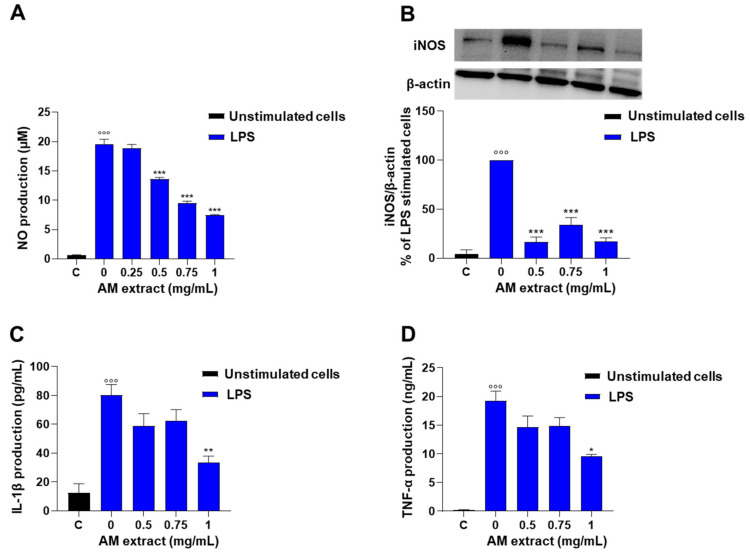
Effect of *A. membranaceus* extract (AM) on nitrite and inflammatory cytokine production in LPS-stimulated murine macrophages: J774 cells were pre-treated for 2 h with AM (0.25–1 mg/mL) prior to LPS stimulation (10 µg/mL) for 24 h. (**A**) Nitrites, stable end-products of NO, were measured in the supernatants by the Griess reaction. (**B**) Western blots analysis with monoclonal antibody to iNOS. iNOS protein was visualized by chemiluminescence Western blotting detection reagents. The bands corresponding to the iNOS protein were quantified by densitometric analysis and the results are expressed as a percentage of LPS-stimulated cells. Densitometric analysis of protein expression represents the mean ± SEM from three separate experiments. Data were normalized on the basis of β-actin. (**C**) IL-1β and (**D**)TNF-α levels were measured by ELISA assay. Data were analyzed by one-way ANOVA plus Bonferroni. Statistical significance is reported as follows °°° *p* < 0.001 vs. unstimulated cells (**C**) and * *p* < 0.05, ** *p* < 0.01 and *** *p* < 0.001 vs. LPS alone.

**Figure 5 ijms-24-10954-f005:**
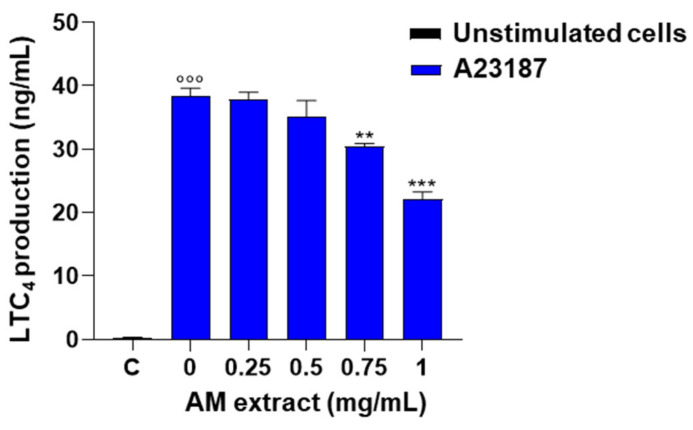
Effect of *A. membranaceus* extract (AM) on LTC_4_ production in mouse peritoneal macrophages. Cells were pre-treated for 30 min with AM (0.25–1 mg/mL) and then stimulated for 60 min with A23187 (0.5 μg/mL). The supernatants were collected for the measurement of LTC_4_ levels by ELISA assay. Values represent means ± S.E.M. (*n* = 3). Data were analyzed by one-way ANOVA plus Bonferroni. Statistical significance is reported as follows °°° *p* < 0.001 vs. unstimulated cells (C); ** *p* < 0.01 and *** *p* < 0.001 vs. A23187 alone.

**Figure 6 ijms-24-10954-f006:**
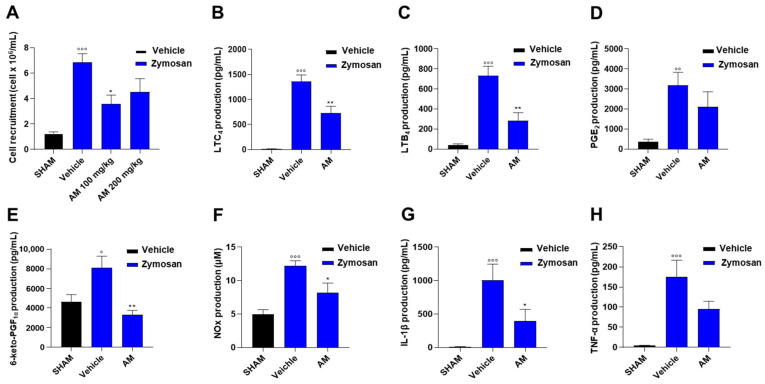
*A. membranaceus* extract (AM) attenuated inflammation in murine peritonitis. Mice received AM (100 mg/kg, i.p.), 30 min before zymosan and were killed 30 min (**B**) or 4 h (**A**–**H**) post peritonitis induction injection. Immune cell infiltration (**A**) in the peritoneal cavity, LTC_4_ (**B**), LTB_4_ (**C**), PGE_2_ (**D**), 6-keto-PGF_1α_ (**E**), IL-1β (**G**) and TNF-α (**H**) levels in the exudate were analyzed by ELISA while NOx (**F**) levels in the exudates were analyzed by Griess assay. Values represent means ± S.E.M. (*n* = 6 mice, each group). Data were analyzed by one-way ANOVA plus Bonferroni. Statistical significance is reported as follows: ° *p* < 0.05, °° *p* < 0.01 and °°° *p* < 0.001 vs. SHAM; * *p* < 0.05 and ** *p* < 0.01 vs. zym + vehicle.

**Figure 7 ijms-24-10954-f007:**
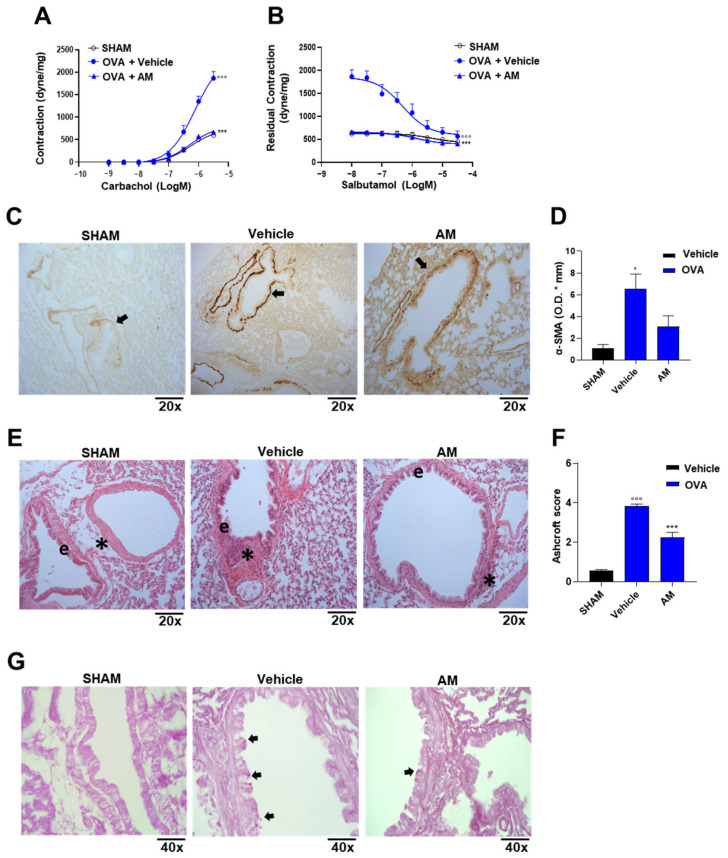
*A. membranaceus* extract (AM) suppresses hallmarks of asthma in mice sensitized to ovalbumin. AM (100 mg/kg) was i.p. administered to mice 30 min prior to injection of ovalbumin (OVA) at days 0 and 7. (**A**) Bronchial reactivity to carbachol or (**B**) salbutamol. (**C**) Immunohistochemical analysis for α-SMA expression on lung slices, arrow: α-SMA positivity. (**D**) Semi-quantitative determination of α-SMA was obtained with ImageJ/Fiji software. (**E**) Lung slices were stained for Hematoxylin & Eosin (H&E; e: bronchial epithelium; *: Peri-bronchovascular interstitium). (**F**) The inflammation was scored for the number and distribution of inflammatory cells on a scale of 0 (no inflammation) to 4 (severe inflammation). (**G**) Periodic acid-Shiff (PAS) staining on lung section (arrows: mucins build-up). The statistical analysis used is two-way ANOVA plus Bonferroni (**A**,**B**) or one-way ANOVA plus Bonferroni post-hoc test (**D**,**F**). Magnification: 20× (**C**,**E**), 40× (**G**). *n* = 3, animals for each group. Statistical significance is reported as follows: ° *p* < 0.05 and °°° *p* < 0.001 vs. SHAM; *** *p* < 0.001 vs. OVA + vehicle.

**Figure 8 ijms-24-10954-f008:**
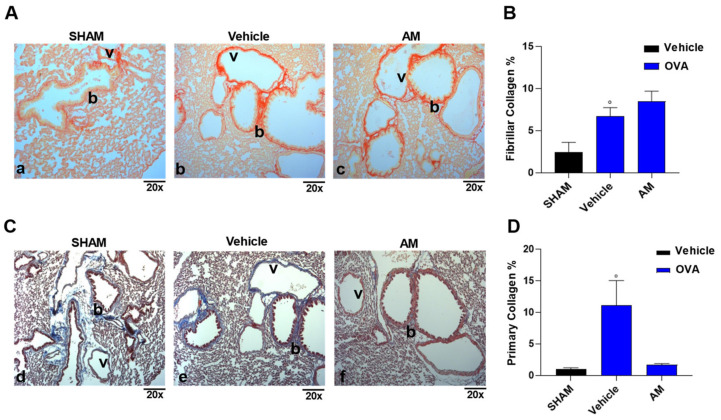
*A. membranaceus* extract (AM, 10 mg/kg) reduces primary collagen in mice sensitized to ovalbumin. Collagen evaluation with Picro-Sirius Red (**A**; **a**–**c**) and Masson’s Trichrome (**C**; **d**–**f**) staining (Magnification: 20×; b = peri-bronchial area; v = peri-vascular area). The percentage of Fibrillar (**B**) and Primary (**D**) collagen was evaluated with ImageJ/Fiji. Statistical analysis used is one-way ANOVA plus Bonferroni post-hoc test (**B**,**D**). *n* = 3, animals for each group. Statistical significance is reported as follows: ° *p* < 0.05 vs. SHAM.

**Figure 9 ijms-24-10954-f009:**
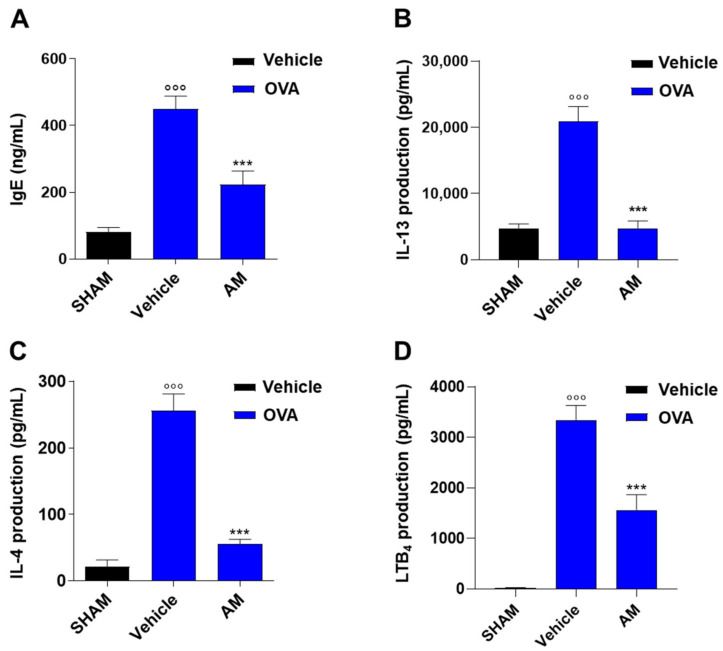
*A. membranaceus* extract (AM, 10 mg/kg) affects the sensitization process induced by allergen exposure. (**A**) Plasma IgE level, pulmonary level of IL-13 (**B**), IL-4 (**C**), and LTB_4_ (**D**). Data were analyzed by one-way ANOVA plus Bonferroni (**A**–**D**). Statistical significance is reported as follows: °°° *p*< 0.001 vs. SHAM; *** *p* < 0.001 vs. OVA + vehicle.

**Figure 10 ijms-24-10954-f010:**
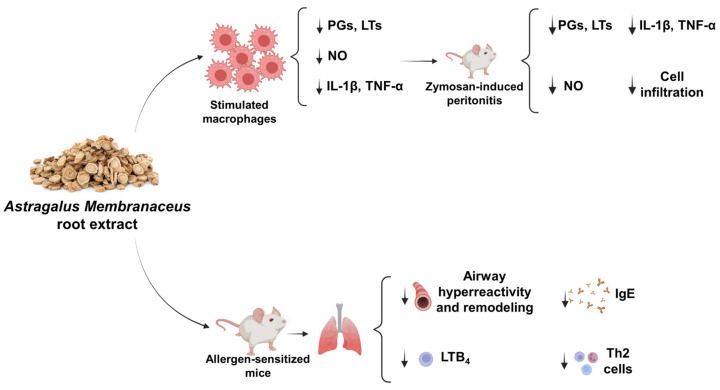
*A. membranaceus* extract inhibits allergic and non-allergic inflammation. Down arrow (↓) = *A. membranceus* inhibitory effect.

**Table 1 ijms-24-10954-t001:** Detected polyphenol of commercial *Astragalus membranaceus* dry extract and relative standard deviation.

Molecule	µg/g	SD (±)
Formononetin	140.35	3.28
Biochanin A	63.17	3.19
Glycitein	30.67	1.81
Diosmetin	11.85	0.30
Hyperoside	9.58	0.38
Chlorogenic acid	7.62	0.36
Apigenin-8-*C*-glucoside (Vitexin)	5.65	0.40
Luteolin	4.96	0.08
Kaempferol	4.77	0.40
Caffeoylquinic acid derivative	4.47	0.11
Ferulic acid	4.29	0.20
Naringenin	4.15	0.22
Pelargonidin	4.01	0.03
Cyanidin	3.56	0.08
Astragalin	3.42	0.12
Delphinidin	3.33	0.15
Naringenin-7-*O*-glucoside	3.27	0.09
Caffeoyliquinic acid lactone	2.91	0.07
Luteolin-6-glucoside (Isoorientin)	2.77	0.11
Quercetin-3-*O*-galactoside	2.75	0.20
Kaempeferol-3-*O*-rhamnoside	2.68	0.13
Eriodictyol	2.67	0.09
Rutin	2.64	0.03
Quercetin-*O*-hexoside	2.63	0.09
Piceatannol	2.57	0.12
Caffeine	2.52	0.07
Myricetin	2.48	0.12
Haesperetin	2.48	0.08
Resveratrol	2.47	0.16
Quercetin	2.46	0.12
Kaempferol-3-*O*-rutinoside	2.46	0.11
Quercetin-*O*-rhamnoside	2.42	0.06
Naringin	2.40	0.14
Linarin	2.32	0.10
Isorhamnetin	2.30	0.09
Quercetin acetylhexoside	2.30	0.05
EGC gallate glucoside	2.30	0.16

## Data Availability

The data presented in this study are available upon request from the corresponding author.

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
