# Peer review of "Beneficial Effects of *Astragalus membranaceus* (Fisch.) Bunge Extract in Controlling Inflammatory Response and Preventing Asthma Features"

_ijms, 2023, doi:10.3390/ijms241310954_

Round 1

Reviewer 1 Report

The manuscript entitled “Beneficial effects of Astragalus membranaceus (Fisch.) Bunge 2 extract in controlling inflammatory response and preventing 3 asthma features” and authored by D’Avino et al demonstrated the anti-inflammatory properties of the commercial extract of A. membranaceus and its useful effects on different asthma-related features. In addition to Astragalus, health-promoting properties of natural products are well-documented. Thus, general background should be added to report that. The following investigations should be considered for integration to cover such benefits: PMID: 33255507, https://doi.org/10.1186/s41936-020-00177-9, PMID: 34639131, https://doi.org/10.4236/ajps.2018.96091.

Other comments

·       Proofreading would be useful.

·       Abbreviations needs to be reviewed.

·       Merge figs 4-6.

·       In figure 2, given that the y-axis represents percentage of cell viability, how come untreated cells show less than 100% whereas all treated cells show much higher cell viability than the 100%?!

·       Original gels of all western blots should be added as supplementary data.

·       What is the purpose of inserts in fig 8G? They seem to have the same magnification and are not discussed in the results.

·       It’d be useful to sketch out a conclusion figure to summarize what overall results of this study.

·       References should be updated.

Needs proofreading

Reviewer 3 Report

Dear Authors,

After reading the manuscript entitled "Beneficial effects of Astragalus membranaceus (Fisch.) Bunge extract in controlling inflammatory response and preventing asthma features", I consider it appropriate to be published in IJMS.

The manuscript is relevant and addresses a very important subject. The abstract clearly summarizes the background results and significance of the study. The introduction presents background information and the aim of this study. Methods presented in the study are well described and overall strategy and analyses are well-reasoned and appropriate to accomplish the specific aims.

A significant negative aspect of the manuscript is the order of the references in the text.

Minor corrections in the manuscript text must be performed to increase its quality:

line 42 is:.. Many studies... – At the end of this sentence, put the appropriate literature in brackets.

line 62 is:.. A. membranaceous.., should be: A. membranaceus – italics

line 77 Table 1 – arrange the molecules from highest concentration to smallest.

line 80 is:.. 0.25-1 mg/mL... – why these concentrations were used in the research? Explain, please.

line 209 Figure 7 contains the abbreviation "SHAM". Please explain what it means.

line 215 is:..OVA…– explain this abbreviation.

line 254 is:.. Figures 9a,b.., should be:...Figure 9A:a,b

line 256 is:.. Figures 9d,e.., should be:...Figure 9C:d,e

line 258 is:.. Figures 9c,D.., should be:...Figure 9A:c and Figure 9D

line 259 is:.. Figures 9f,B.., should be:...Figure 9C:f and Figure 9B

line 284 is:.. that World..., should be:.. that the World..

References:

References [34] and [35] are not cited in the manuscript text.

References [26,27] are cited earlier (line 191) than reference [13] (line 193) - this is incorrect.

Reference [37] is cited earlier (line 431) than reference [36] (line 491) - this is incorrect.
